# 3D Transport Characteristics of Ozone Pollution Affected by Tropical cyclones over the Greater Bay Area, China: Insights from a Radar Wind Profiler Network, Surface observations, and Model Simulations

5

Yuanjian Yang<sup>1</sup>, Chenjie Qian<sup>1</sup>, Minxuan Zhang<sup>1,2</sup>, Chenchao Zhan<sup>1</sup>, Zhenxin Liu<sup>1</sup>, Pak Wai Chan<sup>3</sup>, Xueyan Bi<sup>4</sup>, Meng Gao<sup>2\*</sup>, Simone Lolli<sup>5</sup>

<sup>1</sup>State Key Laboratory of Climate System Prediction and Risk Management, School of Atmospheric Physics, Nanjing
 University of Information Science and Technology, Nanjing 210044, China

<sup>2</sup>Department of Geography, Hong Kong Baptist University, Kowloon Tong, 999077, Hong Kong SAR, China

<sup>3</sup>Hong Kong Observatory, 134A Nathan Road, Kowloon, 999077, Hong Kong, China

<sup>4</sup>Guangzhou Institute of Tropical and Marine Meteorology, China Meteorological Administration, Guangzhou 510080, China

15 <sup>5</sup>CNR-IMAA, Contrada S. Loja, 85050 Tito Scalo (PZ), Italy

Correspondence to: Yuanjian Yang (yyj1985@nuist.edu.cn) or Meng Gao (mmgao2@hkbu.edu.hk)

Abstract. Tropical cyclones (TCs) exert a profound influence on ground-level ozone (O3) pollution dynamics in China's Guangdong-Hong Kong-Macao Greater Bay Area (GBA). Although TC-related O3 transport processes are well recognized, their three-dimensional characteristics remain inadequately characterized. This study provides the first comprehensive observational analysis of O<sub>3</sub> pollution transport mechanisms in the GBA under the influence of TC, integrating high-temporal-resolution wind profile measurements with hourly meteorological and air quality data and model simulations. The findings indicate that TC activity accounts for 39.9% of O<sub>3</sub> pollution episodes in the region, with pollutants advection from northern mainland areas to coastal cities. When TCs are located at a distance of approximately 1800-2000 km, horizontal transport mechanisms dominate, facilitating the conveyance of inland ozone to coastal regions. As the proximity of the TC decreases to within 1000-1700 km, the descending air currents intensify, driving ozone from coastal areas into the boundary layer and resulting in reduced O3 concentrations inland while they increase along the coast. In particular, when TCs approach Taiwan (less than 800 km, NE), increased vertical wind shear occurs about 34.25% than before, particularly over coastal zones, facilitating the injection of freeatmosphere ozone into the boundary layer. This mechanism prolongs surface O3 pollution episodes. Our findings offer critical insights for O<sub>3</sub> pollution mitigation strategies in the GBA and are ofrelevance for other globally significant bay regions susceptible to TC impacts, including Hangzhou Bay (China), Tokyo Bay (Japan), and the Bay of Bengal (India).

### 1 Introduction

30

Tropical cyclones (TCs) are among the most destructive weather systems. However, as they approach, the weather is generally characterized by strong solar radiation, high temperature, and light wind, which is favorable for photochemical production and accumulation of ozone (O<sub>3</sub>) (Parker et al., 2013), which easily causes tropical cyclone-related ozone pollution (TC-O<sub>3</sub>) events (Luo et al., 2018; Yim et al., 2019; Huang et al., 2021). The compound TC-O<sub>3</sub> events have significant climate and environmental impacts, such as worsening air quality (Mills et al., 2018), increasing environmental and health risks (Wang et al., 2020), and affecting socioeconomic activities (Feng et al., 2019). Therefore, it is urgent to clarify the causes of TC-O<sub>3</sub> events to respond appropriately. The Guangdong-Hong Kong-Macao Greater Bay Area (GBA), a region in southern China that covers a total area of 56,000 km2 with a population of more than 70 million, is one of the most economically active zones in the world. In summer and fall, the GBA is susceptible to TCs and has a high O<sub>3</sub> concentration (Li et al., 2019; Liu & Wang, 2020), making it an ideal place to study the TC-O<sub>3</sub> events.

The peripheral circulation of TCs facilitates the generation, accumulation, and transport of O<sub>3</sub> in both vertical and horizontal directions. Due to its distinctive characteristics of the wind field, the three-dimensional transport of pollutants becomes increasingly complex and dynamic with varying times. Vertically, the downdrafts associated with the peripheral circulation of TCs create a warm and dry environment that promotes O<sub>3</sub> formation in the presence of sufficient precursors (Deng et al., 2019). Furthermore, adiabatic warming due to downdrafts can form widespread air stagnation, inhibiting convection (Lolli et al., 2019) and further exacerbating the accumulation of O<sub>3</sub> near the surface (Wei et al., 2016; Zhan et al., 2020; Lin et al., 2024; Shi et al., 2021). Horizontally, strong winds can extend hundreds of kilometers and facilitate long-range transport of O<sub>3</sub> and its precursors, such as nitrogen oxides (NOx) and volatile

https://doi.org/10.5194/egusphere-2025-4668 Preprint. Discussion started: 3 November 2025 © Author(s) 2025. CC BY 4.0 License.

55

65

75

organic compounds (VOC) (Wang et al., 2022; Itahashi, 2023; Xu et al., 2023). As a consequence, O3 pollution occurs. TC dynamic changes in wind fields across different regions not only accelerate the air mass exchange between source and downwind areas but also influence cross-regional air quality, thereby expanding and exacerbating the spread of O<sub>3</sub> pollution. This interregional transport significantly alters the O<sub>3</sub> concentration distribution in inland areas, escalating local pollution events into regional problems. Furthermore, the influence of the peripheral circulation of the TCs on the structure of O<sub>3</sub> is not a static event. Under such meteorological conditions, the processes of O<sub>3</sub> generation, transport and decomposition are interconnected, promoting persistent and spatially extensive pollution. Previous studies on the impact of TCs on increasing O3 concentration were mainly based on numerical simulations, with relatively scarce observational data, especially lacking detailed three-dimensional vertical observations. It is important to note that ozone concentrations in both the troposphere and stratosphere can be influenced during TC-O3 events. Previous studies have shown that tropical cyclone (TC) activity can cause alterations in ozone concentrations within the troposphere (Das et al., 2016; Chen et al., 2021; Li et al., 2021). Additionally, TCs have the potential to transport significant amounts of stratospheric ozone to the surface (Chen et al., 2022), which may exacerbate local O<sub>3</sub> pollution. Furthermore, Li et al. (2020) analyzed 18 years of ozone-sounding data and found that TCs reduce ozone levels in the stratospheric region over the western Pacific, suggesting that TCs can induce variations in ozone concentrations between the troposphere and the tropopause. Hence, enhancing field observations to complement and verify numerical model results is crucial to improving our understanding of the complex mechanisms underlying O<sub>3</sub> transport and formation under the influence of TCs.

Due to the limitations of observational instrumentation, obtaining detailed vertical profiles of atmospheric structure is often problematic. Differences in observational methods and algorithms can also lead to discrepancies in acquired vertical atmospheric information (Guo et al., 2016, 2019; Shi et al., 2020). However, within the boundary layer, where O<sub>3</sub> pollution occurs predominantly, wind profile measurements offer an advantage in terms of high precision and continuity, which are fundamental to accurately characterizing the three-dimensional transport of O<sub>3</sub> (Zhang et al., 2020). Furthermore, the GBA has a relatively dense network of wind profile radars that have been widely applied in air pollution research studies (Jiang et al., 2013; Wu et al., 2015; Liu et al., 2020). By monitoring changes in wind speed, wind direction, and boundary layer height, wind profile radar data provide important empirical support for studying O<sub>3</sub> pollution processes under various weather conditions (e.g., TCs, sea-land wind circulation, wildfires). These data help to improve our understanding of the spatial and temporal distribution characteristics of O<sub>3</sub>, as well as the transport and accumulation processes under complex wind field conditions.

It should be noted that the location peripheral circulation of TC will constantly change with the movement of TC, producing different effects on the atmospheric environment. For example, Huang et al. (2006) found that when a TC is approximately 700 to 1000 km distant from the GBA, the area is susceptible to the influence of the peripheral circulation of the TC, resulting in serious O<sub>3</sub> pollution. Chow et al. (2018) reported that 38.7% of O<sub>3</sub> pollution in Hong Kong occurred when TCs were located close to Taiwan and 58.2% occurred when TCs were located between Taiwan and Hong Kong. Recently, Zhang et al. (2024) found that heatwave events in the GBA are triggered by three TC-related synoptic patterns, and the transformation of these synoptic patterns as TCs move impacts the formation of

heatwave and O<sub>3</sub> pollution in the GBA. Therefore, it is crucial to consider the TC dynamic movement when estimating the influence of its peripheral circulation on O<sub>3</sub> pollution.

TCs generated over the western Pacific Ocean generally follow three main tracks: westward moving, northwestward moving and northeastward recurving. The TC near Taiwan and close to the GBA is promoting TC-O<sub>3</sub> events (Lam et al., 2018; Zhang et al., 2024). This study analyzed O<sub>3</sub> pollution in the GBA and TCs from June to October from 2015 to 2023 to evaluate the relationship between TCs and O<sub>3</sub> pollution. TC Bailu was then selected for an in-depth case study. Based on radar data of the wind profile, air pollutants and meteorological data, this study investigates changes in the structure of the boundary layer under the influence of TC and its impact on O<sub>3</sub> pollution. Two key scientific questions are addressed: (1) What are the characteristics of the winds in the boundary layer affected by the peripheral circulation of the TC as the TC moves? (2) How do changes in winds affect the spatial distribution of O<sub>3</sub> pollution in the GBA? The following sections are organized as follows: Section 2 describes the data and methods. Section 3 presents the main results and discussion, and the conclusions are summarized in Section 4.

#### 2 Materials and Methods

### 2.1 The wind profile radar network

Depending on the TC movement track, three wind profile radar stations along the TC's track within the GBA were selected, namely Huadu (HD; 23.4°N, 113.2°E), Guangzhou (GZ; 22.7°N, 113.5°E) and Hong Kong (HK; 22.3°N, 114.2°E). The locations of these three radars are shown in Figure 1c. Wind profile radars mainly detect the wind field by using the atmospheric turbulence scattering of electromagnetic waves, providing data on horizontal wind direction, horizontal wind speed, vertical wind speed, and atmospheric refractive index structure constant (Cn2) at different heights. These radars have 1-hour averaged data (OOBS) and real-time detection data (ROBS), and the vertical resolution is 60 meters. The detection height of the wind profiler radar is 100 meters in HD and GZ, while it is 300 meters in HK. Furthermore, Cn2 is an important indicator reflecting changes in turbulence intensity, which decreases exponentially with height overall and may have a maximum value or deviate from the normal value at the top of the boundary layer (Angevine et al., 1994; Rb, 2012). This study uses this feature to determine the boundary layer height (BLH).

# 115 2.2 The TC best track dataset

The TC track information in Figure 1b was obtained from the best TC track dataset from the China Meteorological Administration (https://tcdata.typhoon.org.cn/). This dataset includes comprehensive TC tracks in the Northwest Pacific and South China Sea since 1949, with detailed records of the latitude and longitude of the TC, minimum central pressure, and maximum wind speed near the center at a resolution of 6 hours. This dataset provides high precision in coastal and inland areas of the Northwest Pacific region (Ying et al., 2014; Lu et al., 2021). In this study, the TC days were defined as when the TC track enters the region defined by 10°N-30°N latitude and 100°E-130°E longitude. There are three main types of TC tracks in China: the westward tracks (moving from the east of the Philippines westward, often making landfall in Guangdong and Hainan), the northwest tracks (extending northwest, frequently making

landfall in Taiwan, Fujian, and Zhejiang), and the recurving tracks (heading northwest but veering northeast when approaching the eastern coast of China).

#### 2.3 Surface observations

The air quality data in the GBA were sourced from the National Environmental Monitoring Center's National Urban Air Quality Platform (https://air.cnemc.cn:18007/), while data for the HK station were obtained from the Hong Kong Environmental Protection Department's Environmental Protection Interactive Center (https://cd.epic.epd.gov.hk/EPICDI/air/station/). According to the national standard in China, the maximum daily 8hour average (MDA8) O<sub>3</sub> concentration should not exceed 160 µg/m<sup>3</sup>. The O<sub>3</sub> pollution in the GBA is defined as occurring when more than one third of these selected stations record levels of O3 above this threshold. Furthermore, to investigate the spatiotemporal characteristics of pollutants influenced by the TC Bailu, hourly NO2, O<sub>3</sub>, and CO concentration data from the HD, GZ, and HK stations were analyzed due to their proximity to wind profile radar and alignment with the TC Bailu. Meteorological conditions, including 2-meter temperature, 2-meter pressure, 10-meter wind direction and speed, relative humidity, and precipitation, were also assessed for their impact on O<sub>3</sub> concentrations during the TC Bailu event. Meteorological data for the HD and GZ stations were obtained from the China Meteorological Information Center (http://data.cma.cn/), while data for the HK station were gathered from the Hong Kong Observatory (https://www.hko.gov.hk/).

### 140 2.4 Recirculation index

The recirculation index (RI) is calculated to assess the ventilation capacity of the atmosphere and is applicable in situations where pollutants are transported out and then back due to changes in wind direction. It is the cumulative ratio of the vector distance to the cumulative scalar distance of the wind. The formula to determine RI at various heights is expressed as follows Eq. (1) (Zeng et al., 2022; Wu et al., 2015):

$$RI = \frac{\sqrt{\left(\Delta T \Sigma_{k_s}^{k_e} u_k\right)^2 + \left(\Delta T \Sigma_{k_s}^{k_e} v_k\right)^2}}{\Delta T \sum_{k_e}^{k_e} \sqrt{u_k^2 + v_k^2}}, (1)$$

Where, ks is the starting time, ke is the ending time,  $\triangle T$  is the average time interval, generally 24 hours, uk is the radial wind, vk is the zonal wind. RI ranges from 0 to 1. The lower the RI, the worse the horizontal transport capacity of the wind. When RI is close to 1, it indicates significant horizontal transport. In this study, the RI was calculated using wind profiles data and RI=0.6 was selected as a threshold to assess the contribution of pollutants to the GBA return flow. (Chen et al., 2016).

#### 2.5 HYSPLIT model

The Hybrid Single-Particle Lagrangian Integrated Trajectory (HYSPLIT) model is a widely used atmospheric transport model that can calculate the movement trajectories of individual particles or gases in the atmosphere (Lichiheb et al., 2024). It is often used to analyze the transport and diffusion of materials in the past atmospheric environment (Su et al., 2015). The online version is available at https://www.ready.noaa.gov/HYSPLIT traj.php. This

study used the HYSPLIT online backward trajectory module to analyze air mass trajectories at different vertical heights to determine the source of air masses during O<sub>3</sub> pollution events. Heights of 0, 500, 1500, and 2000 m represent the ground, the middle of the boundary layer, the top of the boundary layer, and above the boundary layer, respectively.

#### 2.6 Vertical wind shear

Vertical wind shear (VWS) plays an important role in the dispersion of air pollutants, and thus was calculated here to check its effects on O<sub>3</sub> pollution. The formula is calculated as follows Eq. (2) (Zhang et al., 2020):

$$VWS = \frac{\sqrt{(u_{z1} - u_{z2})^2 + (v_{z1} - v_{z2})^2}}{(z1 - z2)} \times 1000, (2)$$

Where, VWS is the vertical wind shear (units: m·s-1), uz1 and uz2 represent the zonal wind at the height of z1 and z2, respectively; and vz1 and vz2 represent the meridional wind at the height of z1 and z2. z1 is the height above the ceiling and z2 is the height below the ceiling.

#### 2.7 WRF-Chem

This study employed the WRF-Chem (Weather Research and Forecasting model coupled with Chemistry) model with a three-tiered nested grid configuration. The first nested grid (d01) encompassed the majority of Bai Lu's trajectory from formation to landfall, centered at 113.6°E/22.8°N with a horizontal resolution of 27 km. The second and third grids focused on southern coastal China and the Pearl River Delta region, respectively, with resolutions of 9 km and 3 km. Each grid consisted of horizontal dimensions of 103×103, 100×100, and 133×124 cells. The model utilized a Mercator projection with 30 vertical levels that spanned from the surface to 50 hPa.The meteorological initial and boundary conditions for the simulation are derived from the NCEP FNL data, which provide a 1°×1° resolution. Anthropogenic emissions for 2016 at a 0.25°×0.25° spatial resolution were generated by the Tsinghua University's Multi-resolution Emission Inventory for China (MEIC)(Li et al., 2017; Geng et al., 2024). Biomass-related emissions are sourced from the Megan emission inventory. The key parameter settings are detailed in Table 1(Li et al., 2020).

Table 1. Major model configuration options used in the simulations.

| Scheme              | Option                               |
|---------------------|--------------------------------------|
| Microphysics        | Lin scheme                           |
| Longwave radiation  | RRTMG                                |
| Shortwave radiation | Goddard                              |
| Cloud Microphysics  | Grell 3D ensemble scheme             |
| Boundary layer      | Bougeault and Lacarrere (BouLac) PBL |
| Gas-phase chemistry | CBM-Z                                |
| Aerosol chemistry   | MOSAIC-8bins                         |
| Photolysis          | Fast-J photolysis                    |

Figure 1. (a) the simulation domain is illustrated, with three monitoring sites annotated: Huadu (23.4°N, 113.2°E), Guangzhou (22.7°N, 113.5°E), and Hong Kong (22.3°N, 114.2°E). These locations are strategically positioned to capture spatial variations within the study area. (b) Tropical Cyclone Nesat, Lekima, Danas, Bailu. The gray dashed circles are centered on the wind profiler radar in the middle of the GBA at GZ (black dot; 22.7°N,113.55°E), representing distances of 300-2000 km from the GBA, with intervals of 100 km between concentric circles. (c) Locations of the HD, GZ and HK stations in the GBA. The black box indicates the location of the GBA. The black dots indicate the locations of the 10 cities shown in Figure 3.

The validation statistics presented in Table 2 highlight the robustness of the model's performance across key variables. The temperature is simulated with high precision, showing only a minor underestimation of about 1°C. Although wind speed demonstrates a slight overestimation and ozone shows an underestimation, overall agreement between simulated and observed values remains strong. The RMSE values for temperature (1.39), wind speed (1.61), and ozone (36.23) indicate good precision, and the MB values (-1.07 for temperature, +1.48 for wind speed, -3.25 for ozone) reveal consistent biases that are within acceptable ranges for modeling purposes. The FE and FB metrics further validate the model's ability to capture the essential features of these variables. With IOA and R values ranging from 0.24 to 0.64,

the simulation demonstrates a solid foundation for reliable predictions, confirming its applicability for subsequent analyses.

Table 2: Statistical Validation of Meteorological Variables and Ozone in the WRF-Chem Model

|      | Temperature | Wind speed | Wind direction | $O_3$        |
|------|-------------|------------|----------------|--------------|
| RMSE | 1.39 ℃      | 1.61 m s-1 | 80.70°         | 36.23 μg m-3 |
| MB   | -1.07 °C    | 1.48 m s-1 | 63.56°         | -3.25 μg m-3 |
| FE   | -0.04       | 0.75       | 0.29           | 0.48         |
| FB   | -0.04       | 0.74       | 0.33           | -0.06        |
| IOA  | 0.64        | 0.41       |                | 0.24         |
| R    | 0.64        | 0.45       |                |              |

# 3 Results and Discussion

# 200 3.1 Tropical Cyclones and ozone pollution in the GBA

An  $O_3$  pollution day is defined when the maximum daily 8-hour average (MDA8)  $O_3$  concentration exceeds 160  $\mu$ g m $^-$ 3. As shown in Figure 2a, when the TCs move from the northwest Pacific to Taiwan, there were at least two days of  $O_3$  pollution at each point on the grid. The three TC tracks that affect the China traverse regions ( $10^{\circ}$ N $^{\circ}$ N $^{\circ}$ N,  $100^{\circ}$ E $^{\circ}$ 130°E $^{\circ}$ ) contributed to the pollution of  $O_3$  in the GBA. These include the westward tracks, the northwest tracks, and the recurving tracks. Of all the TC track periods in Figure 2a, 408 days were TC days, including 205 days on the westward tracks, 76 days on the northwest tracks, and 127 days on the recurving tracks (Figure 2b). In general, 39.9% of  $O_3$  pollution days in the GBA was associated with TC activity presence, with these tracks contributing 6.5%, 10.6%, and 22.8%, respectively (Figure 2b). These results were consistent with previous studies that indicated that TCs were among the most favorable weather systems for  $O_3$  pollution episodes in the GBA (Jiang et al., 2015; Yang et al., 2019; Lin et al., 2024; Xu et al., 2024).

8



Figure 2. (a) The spatial distribution (in days) of tropical cyclone (TC) activity and ozone (O<sub>3</sub>) pollution days from June to October during the period from 2015 to 2023, at a  $1^{\circ} \times 1^{\circ}$  resolution (unit: day). (b) The number of westward tracks, northwest tracks and westward tracks (blue bar chart, unit: day); probability of O<sub>3</sub> pollution on each track (red bar chart, unit: %).

By examining the MDA8 O<sub>3</sub> concentrations in 10 GBA cities, we observed notable seasonal variations and differences in O<sub>3</sub> levels between cities during June to October (Figure 3). In particular during fall (September to October), there was a tendency for higher O<sub>3</sub> levels, which increased the likelihood of O<sub>3</sub> pollution events. Cities located at higher latitudes, often with mountainous landscapes, generally had higher O<sub>3</sub> concentrations, while those at lower latitudes tended to have lower ones. This pattern was especially evident when TCs affected GBA. Additionally, there was a significant movement of O<sub>3</sub> pollution from inland cities to coastal cities on days influenced by TC activity. During this period, four TCs were selected following three typical tracks for further study to examine the spatial distribution of O<sub>3</sub>. These include Nesat, Bailu (westward track), Danas (northwest track), and Lekima (recurving track) (Figure 3c). The phenomenon also demonstrated the transportation of high-concentration O<sub>3</sub> pollution from inland areas to coastal regions (Figure 4).

Figure 3. The maximum daily 8-hour average (MDA8)  $O_3$  in 10 cities in the GBA in (a) 2015, (b) 2016, (c) 2017, (d) 2018, (e) 2019, (f) 2020, (g) 2021, (h) 2022 and (i) 2023 from June to October. The national standard for ambient air quality for MDA8  $O_3$  is 160  $\mu$ g m $^3$  in China. These cities are sorted by latitude. The black dots represent the days of TC, where the TC track enters (10°N-30°N, 100°E-130°E).

Figure 4. Spatial distribution of surface MDA8 O<sub>3</sub> concentrations in the GBA during (a-c) TC Nasat during 07/26-07/28, 2017, (d-f) TC Danas during 07/16-07/19, 2019, (g-i) TC Lekima during 08/06-08/09, 2019, and (j-l) TC Bailu during 08/22-08/24, 2019.


# 3.2 Spatial characteristics of ozone pollution caused by TC Bailu

During the influence of TC Bailu, the GBA had comprehensive meteorological observations, wind profile radar detections (Figure 2d), and environmental monitoring data, providing a robust data foundation for research. Furthermore, compared to TCs Nasat, Danas, and Lekima, the track of TC Bailu was almost parallel to wind profile radars, allowing better observation of the O<sub>3</sub> transport path. Therefore, we used TC Bailu as a case study to further investigate the mechanisms behind this phenomenon.

TC Bailu formed in the northwest Pacific Ocean (113.8 °E, 13.5 °N) at 16:00 on 20 August 2019, with the intensity of a tropical storm. From 21 to 24 August, Bailu moved northwestward and gradually approached the GBA. The GBA was under the influence of Bailu's peripheral circulation. On 25 August morning, TC Bailu made landfall on the coast of Fujian and its intensity gradually weakened after landfall, eventually dissipating in the afternoon of 26 August. To better understand the relationship between the Bailu TC position and the three wind profile radar stations, Table 3 shows the distances between the TC center and the HD, GZ, and HK stations from the generation phase until before landfall.

Table 3. Distances between the TC center and the HD, GZ, and HK stations, from the generation phase to before landfall.

| Time (LST) | HD   | GZ   | HK   |
|------------|------|------|------|
| 2019082008 | 2429 | 2368 | 2289 |
| 2019082014 | 2393 | 2333 | 2255 |
| 2019082020 | 2330 | 2270 | 2191 |
| 2019082102 | 2263 | 2202 | 2124 |
| 2019082108 | 2214 | 2154 | 2075 |
| 2019082114 | 2103 | 2042 | 1964 |
| 2019082120 | 2021 | 1960 | 1882 |
| 2019082202 | 1939 | 1878 | 1800 |
| 2019082208 | 1876 | 1815 | 1737 |
| 2019082214 | 1828 | 1767 | 1689 |
| 2019082220 | 1737 | 1676 | 1597 |
| 2019082302 | 1601 | 1541 | 1463 |
| 2019082308 | 1490 | 1434 | 1357 |
| 2019082314 | 1348 | 1293 | 1217 |
| 2019082317 | 1292 | 1239 | 1163 |
| 2019082320 | 1241 | 1189 | 1114 |
| 2019082323 | 1167 | 1116 | 1041 |
| 2019082402 | 1070 | 1021 | 947  |
| 2019082405 | 997  | 950  | 877  |
| 2019082408 | 908  | 862  | 790  |
| 2019082411 | 837  | 793  | 723  |
| 2019082414 | 767  | 726  | 658  |
| 2019082417 | 708  | 672  | 607  |
| 2019082420 | 610  | 580  | 520  |






| 2019082423 | 529 | 498 | 439 |
|------------|-----|-----|-----|
| 2019082502 | 499 | 467 | 407 |

Note: Time is displayed as YYYYMMDDHH: Year YYYY, Month MM, Day DD, Hour HH (LST: UTC+8). Distance is measured in kilometers. The TC best track dataset provided position data every 6 hours, and the frequency of the best track updates was increased to every 3 hours for the 24 hours before landfall and during its activity over land in China. Therefore, the distances between the Bailu TCand the wind profile radar at specific times in Figures 5 to 9 are derived from the linear interpolation of the data in the table.

As shown in Figure 5, the O<sub>3</sub> concentration at the HD station was as high as 281 µg m-3 at 15:00 on 22 August, which was related to a higher temperature (35.1 °C), low humidity (47%) and weak winds (2.1 m/s) that promoted the formation and accumulation of O3. On 23 August, the O3 concentration decreased as the temperature decreased. On 24 August, the O<sub>3</sub> concentration decreased significantly. However, the temperature increased and the precursors of O<sub>3</sub> (NO2 and CO) remained almost unchanged. This suggested that O<sub>3</sub> may be transported downwind under the influence of TC Bailu (Figure 5a). The changes in NO2 and CO concentrations were evaluated based on general trends rather than fluctuations observed at a single time point. Although a noticeable minimum in NO2 concentration was recorded on August 24, this does not impact our conclusion regarding the transport of O3 downstream from TC Bailu to the HD station. At the downstream stations, GZ, and HK, O3 concentrations increased from 23 to 24 August. At GZ station, the O<sub>3</sub> concentration even reached 304 µg m-3 on 24 August. Given that NO2 and CO concentrations also increased, the increase in O<sub>3</sub> concentration at the GZ station was the result of both local O<sub>3</sub> formation and regional O<sub>3</sub> transport (Figure 5b). The increase in O3 concentrations resulted from both local O3 formation and regional O3 transport. However, given that TC Bailu had a relatively short time frame from formation to landfall (August 21 - 25), we posit that the concentrations of local O3 and its precursors fluctuated during this period. The significant changes observed in O<sub>3</sub> concentrations were primarily driven by external factors, particularly regional transport of O<sub>3</sub>. Research has shown that the interaction between the TC's outer circulation and large-scale meteorological conditions plays a crucial role in the variation of O3 concentrations, especially under conditions of high temperatures and intense solar radiation (Wang et al., 2024). Regarding the HK station, the NO2 and CO concentrations did not change much. As a developed city, Hong Kong has a high car ownership rate, leading to numerous sources of NOx emissions and high CO levels. This significantly enhances the titration effect of ozone at night, resulting in lower O<sub>3</sub> concentrations on 22 and 23 August. The O<sub>3</sub> concentration on 23 August was slightly lower due to the decrease in temperature. However, on 24 August, the O<sub>3</sub> concentration at the HK station suddenly increased to 361 μg m-3, which was four times higher than on 23 August. Therefore, the high ozone levels recorded at the HK station on August 24 were significantly affected by TC transport. This phenomenon was also confirmed by changes in wind. On 24 August, the dominant wind at the HK station was southwest, which was different from the northwest at the upstream HD and GZ stations. The two air flows converged at the HK station, resulting in severe O<sub>3</sub> pollution at the HK station (Figure 5c).

Figure 5. Time series of wind, air pressure, relative humidity (RH), 24-hour precipitation, CO, NO2, temperature of 2 m (T), and  $O_3$  at (a) HD, (b) GZ, and (c) HK stations from 21 to 25 August 2019. The black boxes indicate the study period.





Figure 6. This figure illustrates the spatiotemporal distribution of ozone concentrations and wind fields simulated by WRF-Chem at different times within the Pearl River Delta region. The stars in the figure denote the locations of three radar stations.

The results of the model corroborate the aforementioned analysis. As shown in Figure 6, there is distinct ozone transport from the northeast to the southern regions during 22–23 July. Additionally, on 24 July, the wind direction near the Hong Kong station changed from northeastto southwest. However, the convergence zone of the south and north winds in the model is located north of the Hong Kong station. This discrepancy arises because, following typhoon landfall, the model slightly overestimates the typhoon's northern position.

In summary, the high  $O_3$  concentrations observed at the HD station from 22 to 23 August were primarily attributed to local photochemical production. On 24 August,  $O_3$  levels at the HD station decreased significantly due to wind changes that led to the transport of  $O_3$  to the downstream region. As a result, elevated  $O_3$  concentrations were observed at the GZ and HK stations, especially at the HK stations where the winds converged.

# 3.3 Transport of ozone pollution affected by TC Bailu

From Section 3.2 above, the regional transport of  $O_3$  induced by the peripheral circulation of TC Bailu was the primary driver of coastal  $O_3$  pollution in the GBA. In this section, we used high resolution boundary layer observation data to describe the three-dimensional transport of  $O_3$ , both in the horizontal and vertical dimensions.

### 3.3.1 Horizontal transport and accumulation by recirculation

Based on the 24-hour backward trajectories, the air masses within the boundary layer circulated around the HD station, while the air masses above the boundary layer originated from the northeast (7a-c). In terms of GZ and HK stations, air masses above the boundary layer on 22 to 23 August also came from the northeast, while air masses within the boundary layer came from the South China Sea region (Figure 7d-h). Then on 24 August, the surface and 500-m air mass transport pathways crossed HD and GZ stations, advecting O<sub>3</sub> from HD to GZ and then to HK (Figure 7f and i). Based on the distance from TC Bailu to the wind profile radars, when this distance was approximately 1600 to 1800 km, an influx of air masses from other regions occurred in the high-altitude layers of the GBA, persisting until TC Bailu made landfall.



Figure 7. 24-hour backward trajectories of (a-c) HD station, (d-f) GZ station, and (g-i) HK station at different vertical heights. (Heights of 0, 500, 1500, and 2000 m represent the surface, the middle of the boundary layer, the top of the boundary layer, and above the boundary layer, respectively. The distance in the character brackets at the top left of the subplot indicated the distance from TC Bailu to the wind profile radars at that moment.)

Figure 8 further illustrated the vertical profiles of the horizontal wind at these three stations. On 23 August, as TC Bailu approached (about 1000-1600 km away from the GBA), the wind speed in and above the boundary layer increased, and the wind speed above the boundary layer increased even more. When Bailu was located near Taiwan on 24 August (15:00, approximately 600-700 km from the GBA), the wind speed continued to increase, but the wind direction in the upper boundary layer shifted from northeast to north or northwest (Figure 8a). Regarding the GZ and HK stations, the wind speed in the boundary layer gradually strengthened and deepened from 24 August, accompanied by an increase in O<sub>3</sub> concentrations. Additionally, the height of the boundary layer in the GBA was lower compared to the previous two days, indicating a more stable atmospheric boundary structure that was unfavorable for the dispersion of pollutants (Figure 8b-c).



Figure 8. Horizontal wind (vector arrows) measured by wind profile radar at the (a) HD, (b) GZ, and (c) HK stations from 22 to 25 August. Black dashed and blue lines represent the height of the boundary layer and the concentration of O<sub>3</sub>, respectively. The distance in the brackets below the date on the x-axis representd the distance from TC Bailu to the wind profile radars at that moment.

Since the direction of the wind in the GBA changed significantly due to TC Bailu, it was possible that the dispersed O<sub>3</sub> was transported back to the local area, prolonging the O<sub>3</sub> pollution. In this study, the Recirculation Index (RI) was used to evaluate the effect of wind on re-accumulation of O<sub>3</sub>. As illustrated in Figure 9, at the three stations, the RI was less than 0.6 within the boundary layer from August 22 to 23 and greater than 0.6 outside the boundary layer, when TC Bailu was approximately 1800-2000 km from the GBA. This configuration could have trapped O<sub>3</sub> into the surrounding areas. On 24 August (about 500-1000 km away from the GBA), the RI increased and the O<sub>3</sub> concentration decreased at the HD station, while the opposite was observed at the GZ and HK stations. In particular, at the HK station, the surface wind direction changed from southeast (21 August) to westerly (22 and 23 August) and then to strong northwest (24 August). The near-surface convergence effect trapped O<sub>3</sub> near the surface even though the diffusion conditions aloft were favorable (Figure 5c). Additionally, it is important to note that the scanning height of the HK radar is 300 meters. Consequently, when analyzing ground-level O<sub>3</sub> concentrations in Hong Kong, we primarily rely on the wind direction reported by the local meteorological station for our assessments and analyses.

Although favorable high-altitude dispersion conditions can facilitate the spread of O<sub>3</sub>, excessive convergence effects near the surface can create a 'bottleneck', trapping O<sub>3</sub> close to the ground and inhibiting its upward movement.

Figure 9. The recirculation index (shaded) was calculated using the wind profile radar at (a) the HD station, (b) GZ station, and (c) HK station from 22 to 25 August. Black dashed and blue lines represent the height of the boundary layer and the concentration of O<sub>3</sub>, respectively. The distance in the brackets below the date on the x-axis represents the distance from TC Bailu to the wind profile radars at that moment.


Figure 10. The figure displays time series of the ozone and wind profile simulated between 00:00 on August 22 and 00:00 on August 25 at three sites (Huadu, Guangzhou, and Hong Kong). Vertical wind speed has been amplified by a factor of 10 to enhance clarity. In the figure, the red line indicates the boundary layer height.

Figure 10 shows that on 23 August at midnight, when the typhoon was still distant from the Pearl River Delta region, subsiding flow occurred at the boundary layer height across all three sites. This facilitated notable transport of ozone from the upper boundary layer to the lower part of the layer. However, after approximately 12:00 PM on August 23, the near-surface recirculation index increased at all three locations, with horizontal advection intensifying. Although ozone concentrations at higher altitudes remained largely unchanged, a significant low-value zone formed below the boundary layer in all regions, likely caused by ozone dispersion due to horizontal wind conditions.

## 3.3.2 Entrainment and vertical mixing in the boundary layer

The convergence and divergence of the horizontal wind influences the vertical wind speed. Updrafts (convergence) can transport heat and O<sub>3</sub> from the lower layers to higher layers, while downdrafts (divergence) the opposite. Figure 11 shows the vertical wind at the HD station, GZ station and the HK station during period under study. From 22 to 23 August (about 1000-2000 km away from the GBA), all three stations had strong downdrafts within the boundary layer, especially during the O<sub>3</sub> peak. However, on 24 August (14:00, 767 km from the GBA), updrafts appeared at the HD


station (Figure 11a), which transported O<sub>3</sub> upward. At GZ and HK stations, there were still strong downdrafts within 3 km (Figure 11b-c). On 24-25 August, the differences in RI in GZ indicated that the O<sub>3</sub> concentration within the boundary layer was significantly influenced by local convergence effects, resulting in the accumulation of O<sub>3</sub> near the surface. This phenomenon may be attributed to the topography and level of urbanization in GZ, which led to reduced wind speeds within the boundary layer, thus limiting the vertical mixing of O<sub>3</sub>. On the contrary, the vertical wind speeds inside and outside the boundary layer in HK were nearly identical, suggesting a more uniform dynamic structure that facilitated the vertical mixing of O<sub>3</sub>. In this case, O<sub>3</sub> transported horizontally from the upstream could be carried to the surface.

Figure 11. Vertical wind speed (color filled) was measured by wind profile radar at the (a) HD, (b) GZ and (c) HK stations from 22 to 25 August. The black dashed and blue lines represent the height of the boundary layer and the concentration of O<sub>3</sub>, respectively. The distance in brackets below the date on the x-axis represents the distance from TC Bailu to the wind profile radars at that time

On the other hand, VWS can accelerate the vertical mixing of  $O_3$  in the boundary layer. As shown in Figure 12, on 22 August (15:00, about 1600-1800 km away from the GBA), the VWS centers were visible around 1.5 km and 2.5 km,

with a maximum of 70 m·s-1at the GZ station. On 23 August (15:00, about 1000-1300 km away from the GBA), the HD station had a more consistent wind direction in the upper part of the boundary layer (Figure 8a), and the VWS was relatively small (Figure 12a). However, the wind direction was more complex at the GZ and HK stations in the upper part of the boundary layer (Figure 8b-c), leading to increased VWS with values around 7 m·s-1 within the boundary layer (Figure 12b-c). On 24 August (15:00, about 600-800 km away from the GBA), obvious VWS appeared within the boundary layer at all three stations. VWS centers were most pronounced around 1 km at the HD and GZ stations. But the HK station had a higher boundary layer height that could accommodate more O<sub>3</sub>.

Figure 12. Vertical Wind Shear (VWS) from 22 to 24 August at (a-c) HD station, (d-f) GZ station and (g-i) HK station. The black dashed line represents the height of the boundary layer. The distance in character brackets at the top left of the subplot indicated the distance from TC Bailu to the wind profile radars at that moment.





Figure 13. This normalized time series shows the contributions of three dynamical processes (horizontal transport, vertical transport and turbulent mixing) to ozone pollution in the Pearl River Delta region. The dashed lines divide the processes into three distinct periods: a period dominated by horizontal and descending transport (Period I), a period with enhanced descending transport (Period II), and a period characterized by increased turbulent mixing (Period III).

In this study, we extracted the advection term (advh) for ozone, the vertical transport term (advz), and the turbulent mixing term (vmix) from the WRF-Chem model, and set all negative contributions of ozone formation within the Pearl River Delta region to zero, considering only positive contributions. According to Eq. (3):

$$P_i^+ = \frac{C_i^+}{C_{total}^+} \times 100\%$$
, (3)

where  $C_i^+$  represents the sum of positive contributions from a specific process among the three processes within the Pearl River Delta region, and  $C_{total}^+$  denotes the total positive contributions from all three processes. The ratio is used to quantify the contribution of each dynamic process to ozone pollution.

Given that horizontal and vertical transport dominate while turbulent mixing plays an auxiliary role, we performed normalization on the contribution rates of the three processes to analyze their trends. We observed significant fluctuations in the contribution rates of horizontal and vertical transport between 22 August and 20:00 on 23 August. From 20:00 on 20 August to 04:00 on 25 August, vertical transport strengthened and stabilized, while horizontal transport weakened and stabilized. Between 04:00 and 20:00 on 25 August, the contribution rate of turbulent mixing increased dramatically (by over 34.15%), accompanied by a sharp decline in vertical transport contributions and a significant increase in horizontal transport contributions. By 20:00 on 25 August, turbulent mixing contributions stabilized, while vertical transport contributions gradually increased and horizontal transport contributions decreased. We calculated the average contribution rates of the three processes during three key stages (horizontal and vertical transport dominance period, enhanced vertical transport period, and enhanced turbulent mixing period), as shown in the table below.



Table 4: Contribution rates of three transport processes averaged across three key periods.

|                      | Phase 1 | Phase 2 | Phase 3 |
|----------------------|---------|---------|---------|
| Horizontal transport | 54.26%  | 53.81%  | 54.11%  |
| Descending transport | 45.37%  | 45.79%  | 45.34%  |
| Turbulent Mixing     | 0.37%   | 0.41%   | 0.55%   |

We selected the areas of Huadu, Guangzhou, and Hong Kong as representative centers for the Pearl River Delta region and analyzed the relationship between typhoon distance from the regional center and its contribution rate. The results show that the influence of the typhoon on the region exhibits a significant distance dependence: there is a certain correlation between the contribution rate and the inverse of the distance. The specific fitting formulas are as follows:

$$Vmix : y = \frac{88.80}{x} + 0.28 \qquad R = 0.86, (4)$$

$$Advz : y = \frac{-159.48}{x} + 45.84 \quad R = 0.61, (5)$$

$$Advh : y = \frac{70.68}{x} + 53.88 \qquad R = 0.37, (6)$$

Of particular note is the significant inverse relationship between turbulent mixing contribution and typhoon distance from the regional center. When the typhoon is far from the Pearl River Delta, its contribution to turbulent mixing is relatively low; as the typhoon approaches the region, the contribution rate increases gradually. However, when the typhoon is near the region, the growth rate of the contribution rate significantly accelerates, indicating that turbulent mixing has a more pronounced effect on the Pearl River Delta when the typhoon is at close range.

# 3.3.3 A schematic diagram of 3-D ozone transport affected by TCs

This study dynamically revealed how the peripheral circulation of TC changed the higher O<sub>3</sub> concentration in the GBA region. There were three main phases. The first phase was dominated by the downdrafts. When TC was closed enough (about 1800—2000 km), the GBA was primarily controlled by horizontal and descending airflow(Figure 14a), horizontal airflow transported inland O<sub>3</sub> towards coastal areas. The second phase involved the descending airflow. As TC approached (about 1000—1800 km), subsidence in the PBL facilitated downward transport of upper-layer O<sub>3</sub> into the PBL. This caused reduced surface O<sub>3</sub> concentrations over inland regions due to horizontal dispersion, weakening positive contributions. Furthermore, the change in the direction of the surface wind then caused the O<sub>3</sub> to recirculate and accumulate near the surface, especially for coastal areas (Figure 14b). The third phase demonstrated vertical mixing. As the TC moved near Taiwan (less than 800 km), horizontal transport of O<sub>3</sub> began to strengthen. Furthermore,

entrainment and strong vertical mixing led to an increase in O<sub>3</sub> concentration at the surface (Figure 14c). Li et al. (2022) also found that airflow on the periphery of TC can enhance vertical mixing of O<sub>3</sub>. Our results confirmed this statement and provide first observational evidence for a full picture of the transport of O<sub>3</sub> pollution due to TCs within the boundary layer.

Figure 14. A schematic diagram illustrating the dynamic response of airflow and O3 within the height of the boundary layer height (BLH) of the GBA to the approach of a tropical cyclone (TC). (a) Phase I: Horizontal & Downdraft Control - The pink arrows represent the downdraft airflow from the TC's periphery. The orange arrows indicate the northeasterly winds that transport O3 horizontally across the GBA. (b) Phase II: Descending Airflow (c) Phase III: Vertical Mixing - The black arrows depict the vertical mixing of air and O3 within the boundary layer.

### 4 Conclusions

- O<sub>3</sub> pollution in the GBA is closely related to TC. However, the characteristics of the boundary layer winds influenced by the peripheral circulation of TCs and their impacts on spatial variations in O<sub>3</sub> remain unclear. This study comprehensively examined this problem using vertical observations from the wind profile radar combined with surface O<sub>3</sub> and meteorological observations. The main findings are summarized below:
- TC activity is responsible for 39.9% of O<sub>3</sub> pollution days in the GBA. With TCs, the spatial evolution of O<sub>3</sub> pollution is marked by its transport from inland cities to coastal cities. The transport process can be roughly divided into three phases, depending on the dominant factors (downdrafts, horizontal wind, and vertical mixing):
  - 1) When TC was closed enough (about 1800—2000 km), the GBA was primarily controlled by horizontal and descending airflow, horizontal airflow transported inland O<sub>3</sub> towards coastal areas.
- 2) As TC approached (about 1000—1800 km), the subsidence of the planetary boundary layer (PBL) enhanced the downward transport of upper-level O<sub>3</sub> into the PBL. This process led to a decrease in surface O<sub>3</sub> concentrations in inland areas due to horizontal dispersion, weakening the positive contributions to surface O<sub>3</sub> levels.
  - 3) As TC moved near Taiwan (less than 800km), the horizontal transport of O<sub>3</sub> increased. simultaneously, vertical wind shear increased significantly, capturing O<sub>3</sub> in the boundary layer and improving mixing, causing O<sub>3</sub> pollution to move to coastal areas.
- Our work provides the first observational evidence of O<sub>3</sub> pollution in the GBA affected by the peripheral circulation of TC, presenting a three-dimensional perspective of O<sub>3</sub> transport in the boundary layer. These results can be applied to other regions susceptible to TC and contribute to local O<sub>3</sub> pollution prevention strategies. However, we acknowledge insufficient research on the other two typical TC paths (northwest track and recurving track). Future studies should conduct more in-depth analyses of these paths to fully understand the impact of different TC tracks on O<sub>3</sub> pollution.

# Financial support

This research was supported by the National Natural Science Foundation of China (42175098, 42222503 and 42201053) and supported by a State Key Laboratory of Resources and Environmental Information System grant. The authors gratefully acknowledge the Hong Kong Observatory for providing the essential data used in this research.

### **Author contributions**

YY: Conceptualization, Methodology, Investigation, Formal Analysis, Writing-Original Draft, Writing – Review & Editing, Supervision, Project Administration; CQ: Investigation Data Curation, Simulation, Visualization. MZ: Investigation, Resources. CZ: Investigation, Software. ZL: Investigation, Resources. XB: Investigation, Resources; PC:Writing – Review & Editing; GM: Methodology, Writing – Review & Editing.

# **Competing interests**

At least one of the (co-)authors is a member of the editorial board of Atmospheric Measurement Techniques.

### 495 Data Availability Statement

All the data used in this paper are publicly available. The datasets that are analyzed and used to support the findings of this study are available in public domain. Hourly meteorological data and wind profile radar data can be obtained from the National Meteorological Information Center of the China Meteorological Administration (CMA, http://data.cma.cn/). The HK station meteorological data was obtained from the Hong Kong Observatory (https://www.hko.gov.hk/). The hourly NO2, O3 and CO concentrations at ground level for the HD and GZ stations are obtained from the National Environmental Monitoring Center's national urban air quality platform (https://air.cnemc.cn:18007/), while data for the HK station were obtained from the Hong Kong Environmental Protection Department's Environmental Protection Interactive Center (https://cd.epic.epd.gov.hk/EPICDI/air/station/). The CMA released the best tropical cyclone track data set (https://tcdata.typhoon.org.cn/).

### 505

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
