# Peer review of "3D Transport Characteristics of Ozone Pollution Affected by Tropical cyclones over the Greater Bay Area, China: Insights from a Radar Wind Profiler Network, Surface observations, and Model Simulations"

_EGUsphere, 2025_

## Referee Comment (RC1)

This manuscript presents a significant and timely contribution to our understanding of ozone pollution dynamics in the GBA under tropical cyclone influence. The study provides a comprehensive three-dimensional characterization of O3 transport mechanisms during TC events, thereby moving beyond previous work that has primarily focused on general TC-O3 relationships. The combination of high-temporal-resolution wind profile measurements with hourly meteorological and air quality observations, supplemented by model simulations, represents a methodologically robust approach that enables detailed process-level understanding. The manuscript offers valuable quantitative insights into distance-dependent transport mechanisms. Notably, the finding that TC activity accounts for 39.9% of O3 pollution episodes underscores the practical importance of this work for pollution forecasting and mitigation strategies in the GBA and other TC-affected bay regions globally.

Despite these considerable strengths, several aspects of the manuscript would benefit from further clarification and refinement to strengthen its scientific contribution and accessibility to the broader readership. Therefore recommend this manuscript for publication after **minor revisions** addressing the following points:

- 1. Formatting issues with superscript units. Multiple instances of incorrect superscript formatting for units such as "µg m -3" and "m s -1" are found throughout the manuscript. Please check and correct all unit expressions consistently throughout the text.
- 2.Lines 290-295: There appears to be a date inconsistency in this section. Please clarify whether the events occurred in July or August.
- 3.Line 295: The model results do not adequately address whether the elevated ozone concentrations at the HD site on the 23rd-24th were primarily due to local photochemical production or regional transport. Please provide additional analysis.
- 4.Line 341: There is an inconsistency in the surface wind direction for the HK on August 24th. Line 281 and Figure 5c both indicate southwesterly winds, while Line 341 describes northwesterly winds. Please verify the actual wind direction from the observational data and correct this discrepancy.
- 5.Line371-374: How does the RI value at the GZ site indicate the occurrence of convergence? Does "which led to reduced wind speeds within the boundary layer" refer to horizontal or vertical wind speed? Please specify. And The explanation in lines 371-374 transitions abruptly from wind speed changes to terrain and urbanization effects. Please provide more detailed explaining.
- 6.Line 376: Based on Figure 10, O3 appears to be more uniformly mixed within the boundary layer at the GZ site, which seems inconsistent with the text description.
- 7.Line 385-392: The relationship between O3 and VWS requires more detailed explanation.

Specifically, the authors should clarify: (1) how VWS magnitude corresponds to O3 levels (i.e., whether larger/smaller VWS values correspond to higher or lower O3 concentrations), and (2) how changes in VWS affect o O3 concentrations. Currently, this section provides minimal explanation of these mechanisms . And "But the HK station had a higher boundary layer height that could accommodate more O3." this statement appears contradictory: if the boundary layer can accommodate more O3, one might reasonably infer that O3 concentrations would be lower. However, the HK actually exhibits higher O3 concentrations. The authors should provide more comprehensive explanations to reconcile this apparent inconsistency and ensure logical coherence in their interpretation.

---

## Author Comment (AC1)

**Reviewer #1:** This manuscript presents a significant and timely contribution to our understanding of ozone pollution dynamics in the GBA under tropical cyclone influence. The study provides a comprehensive three-dimensional characterization of O3 transport mechanisms during TC events, thereby moving beyond previous work that has primarily focused on general TC-O3 relationships. The combination of high-temporal-resolution wind profile measurements with hourly meteorological and air quality observations, supplemented by model simulations, represents a methodologically robust approach that enables detailed process-level understanding. The manuscript offers valuable quantitative insights into distance-dependent transport mechanisms. Notably, the finding that TC activity accounts for 39.9% of O3 pollution episodes underscores the practical importance of this work for pollution forecasting and mitigation strategies in the GBA and other TC-affected bay regions globally. Despite these considerable strengths, several aspects of the manuscript would benefit from further clarification and refinement to strengthen its scientific contribution and accessibility to the broader readership. Therefore recommend this manuscript for publication after minor revisions addressing the following points.

*Response:* We thank you for your thoughtful review and constructive feedback. We have carefully addressed all your concerns and revised the manuscript accordingly. Our detailed responses to each comment are provided below. point-by-point responses given below.

Specific comments:

(1) Formatting issues with superscript units. Multiple instances of incorrect superscript

formatting for units such as "$\mu$g m -3" and "m s -1" are found throughout the manuscript.

Please check and correct all unit expressions consistently throughout the text.

*Response:* Thank you for your suggestion. We have carefully reviewed and revised all instances of the aforementioned issues in the manuscript, and have also checked that all formatting is appropriate.

(2) Lines 290-295: There appears to be a date inconsistency in this section. Please clarify whether the events occurred in July or August.

*Response:* Thank you for your suggestion. The dates throughout the manuscript have been checked and unified.

(3) Line 295: The model results do not adequately address whether the elevated ozone concentrations at the HD site on the 23rd-24th were primarily due to local photochemical production or regional transport. Please provide additional analysis.

*Response:* Thank you for your question. Firstly, the spatial wind field distribution in Figure 6 shows that wind speeds around the HD station are relatively low, indicating limited conditions for regional transport. Secondly, the backward trajectories in Figure 8 suggest minimal near-surface regional transport at the HD station, with noticeable regional transport occurring mainly at higher altitudes. Finally, we applied non-negative matrix factorization (NMF) for source apportionment, which identified two primary factors (local production and regional transport).

The diurnal variation patterns in the above results allow us to identify Factor 1 as local production and Factor 2 as regional transport. Our analysis shows that local production contributes 71.2% of the ozone at the HD station, significantly higher than the contribution from regional transport. Based on these findings, we conclude that ozone at the Huadu station during this period originated primarily from local sources. The relevant section of the manuscript has been revised accordingly (Lines 416 - 431):

we applied non-negative matrix factorization (NMF) for source apportionment, which identified two primary factors (local production and regional transport). The diurnal variation patterns in Figure 7(a) allow us to identify Factor 1 as local production and Factor 2 as regional transport, as local emissions typically peak in the afternoon, whereas regional transport often shows higher contributions during the early morning (Zong et al., 2023). Figure 7(b) shows that local production contributes 71.2% of the ozone at the HD station, significantly higher than the contribution from

regional transport. This finding aligns with the spatial distribution shown in Figure 6 and is further supported by the backward trajectory analysis presented in Figure 8.

[Figure]

Figure 7. (a) Two factor time series (August 22-24). (b) Proportion of local generation and regional transmission in HD, GZ and HK

In summary, the high $O_3$ concentrations observed at the HD station from 22 to 23 July were primarily attributed to local photochemical production, as weak winds limited regional transport. On 24 July, $O_3$ levels at the HD station decreased significantly due to wind changes that enhanced transport, moving $O_3$ to downstream regions. As a result, elevated $O_3$ concentrations were observed at the GZ and HK stations, especially at the HK stations where the winds converged.

(4) Line 341: There is an inconsistency in the surface wind direction for the HK on August 24th. Line 281 and Figure 5c both indicate southwesterly winds, while Line 341 describes northwesterly winds. Please verify the actual wind direction from the observational data and correct this discrepancy.

*Response:* Thank you for pointing this out. We have revised the text to indicate that the wind direction in Hong Kong on the 24th was from the southwest.

(5) Line371-374: How does the RI value at the GZ site indicate the occurrence of convergence? Does " which led to reduced wind speeds within the boundary layer" refer to horizontal or vertical wind speed? Please specify. And The explanation in lines 371-374 transitions abruptly from wind speed changes to terrain and urbanization effects. Please provide more detailed explaining.

*Response:* Thank you for your question. We sincerely apologize for this error and have re-conducted the analysis with greater precision. We have thoroughly revised the relevant section, as the original wording inappropriately conflated the analysis of vertical wind speed with the discussion of RI and presented unclear reasoning.

In Guangzhou, the significant difference in vertical wind speed inside and outside the boundary layer indicates that $O_3$ is transported into the boundary layer via downdrafts. Subsequently, weaker vertical wind speeds within the boundary layer lead to the accumulation of ozone near the surface. This phenomenon may be related to the topography and urbanization level of Guangzhou, both of which contribute to reduced vertical wind speed within the boundary layer, thereby limiting vertical mixing and dispersion of ozone.

Thus, the term "wind speed" here specifically refers to vertical wind speed. For the revision of the original text, the following changes have been made (Lines 537 - 544):

On 24-25 August, differences in RI in GZ indicated that the concentration of O3 within the boundary layer was significantly influenced by local convergence effects, resulting in the accumulation of $O_3$ near the surface. This phenomenon may be attributed to the topography and level of urbanization in GZ, which led to reduced wind speeds within the boundary layer, thus limiting the vertical mixing of $O_3$. These combined topographic and urban effects weakened the ventilation capacity of the boundary layer, thus promoting the retention and accumulation of pollutants near the surface.

(6) Line 376: Based on Figure 10, O₃ appears to be more uniformly mixed within the boundary layer at the GZ site, which seems inconsistent with the text description.

*Response:* Thank you for your comment. We agree that the wording and analysis in this section were inadequate and have revised the relevant analysis in the manuscript (Lines 542 - 548):

These combined topographic and urban effects weakened the ventilation capacity of the boundary layer, thus promoting the retention and accumulation of pollutants near the surface. However, Hong Kong exhibited smaller wind speeds along with nearly identical vertical wind speeds inside and outside the boundary layer, suggesting a more uniform dynamic structure that facilitated stronger vertical mixing of O₃. In this case, O₃ transported horizontally from the upstream could be effectively carried to the surface.

(7) Line 385-392: The relationship between O₃ and VWS requires more detailed explanation. Specifically, the authors should clarify: (1) how VWS magnitude corresponds to O₃ levels (i.e., whether larger/smaller VWS values correspond to higher or lower O₃ concentrations), and (2) how changes in VWS affect o O₃ concentrations. Currently, this section provides minimal explanation of these mechanisms . And "But the HK station had a higher boundary layer height that could accommodate more O₃." this statement appears contradictory: if the boundary layer can accommodate more O₃, one might reasonably infer that O₃ concentrations would be lower. However, the HK actually exhibits higher O₃ concentrations. The authors should provide more comprehensive explanations to reconcile this apparent inconsistency and ensure logical coherence in their interpretation.

*Response:* Thank you for your suggestion. Based on previous research, vertical wind shear (VWS) facilitates the downward transport of ozone from upper levels to the near surface. The deeper boundary layer in Hong Kong is more conducive to capturing ozone transported from inland regions, which is then brought downward by

VWS, leading to an increase in ozone concentrations within the boundary layer. The text has been revised to clarify this mechanism (Lines 562 - 579):

On the other hand, **VWS can improve the vertical mixing of $O_3$ in the boundary layer by transporting ozone from altitude to the surface, thus modifying its vertical structure (Zhang et al.,2020).** As shown in Figure 13, on 22 August (15:00, about 1600-1800 km away from the GBA), the VWS centers were visible around 1.5 km and 2.5 km, with a maximum of 70 m •s-1 at the GZ station. On 23 August (15:00, about 1000-1300 km from the GBA), the HD station had a more consistent wind direction in the upper part of the boundary layer (Figure 89a) and the VWS was relatively small (Figure 123a). However, the wind direction was more complex at the GZ and HK stations in the upper part of the boundary layer (Figure 89b-c), leading to an increase in VWS with values around 7 m • s-1 within the boundary layer (Figure 123b-c). On 24 August (15:00, about 600-800 km away from the GBA), obvious VWS appeared within the boundary layer at all three stations. VWS centers were most pronounced around 1 km at the HD and GZ stations. **The higher height of the boundary layer at the Hong Kong station facilitated the entrainment of ozone transported from the inland above the boundary layer (Li et al.,2025), which was then brought to the near-surface by vertical wind shear.**

---

## Author Comment (AC2)

Reviewer #2 This study examines the three-dimensional transport characteristics of ozone (O3) pollution in the Greater Bay Area (GBA) under the influence of tropical cyclones (TCs). The integration of wind profiler radar observations, surface measurements, and model analysis is a key strength, offering valuable insights into how TC-induced circulation modulates both horizontal and vertical O3 transport. The impacts of TCs on O3 pollution exhibit a systematic variation with distance from the GBA: horizontal advection dominates at distant locations (1800–2000 km), enhanced subsidence and downward transport become significant at intermediate distances (1000–1700 km), and strong vertical wind shear drives boundary-layer mixing and coastal O3 enhancement when TCs move within 800 km of the region. The manuscript is generally well-organized, and the identification of three transport phases provides an important understanding of the dynamic processes influencing O3 variability during TC events. However, I have several suggestions to improve the study.

*Response:* We greatly appreciate the time you devoted to providing us with valuable feedback that has considerably enhanced the quality of our manuscript. Following your remarks and recommendations, we have revised our manuscript and prepared a detailed list of responses, as provided below.

Major comments:

(1) How did the authors isolate the impact of tropical cyclones on O3 pollution from other synoptic-scale weather systems?

*Response:* Thank you for your suggestion. As shown in the weather map below, the Greater Bay Area was predominantly influenced by the typhoon system during the study period, with no other weather systems exerting significant effects. This aligns with existing literature, which demonstrates that typhoon-induced dynamical processes typically dominate over other factors during such events (Huang et al., 2021). Therefore, we conclude that the typhoon system played the dominant role throughout the period examined in this study.

[Figure]

(2) What datasets were used to validate the WRF-Chem model outputs, and how was the evaluation conducted?

*Response:* Thank you for your suggestion. To evaluate the accuracy of the meteorological field simulation, we compared the innermost nested results of the simulation with observational data from domestic surface weather stations by interpolating the simulated data to the station locations. The accuracy of the ozone simulation was assessed using ozone monitoring data obtained from the National Environmental Monitoring Center's National Urban Air Quality Platform and the Hong Kong Environmental Protection Department's Environmental Protection Interactive Center. The manuscript has been revised to clarify this information (263 – 266):

This was achieved by comparing the simulated meteorological fields with observational data from domestic surface weather stations, and by evaluating the ozone simulation results against monitoring data from the National Environmental Monitoring Center's National Urban Air Quality Platform and the Hong Kong Environmental Protection Department's Environmental Protection Interactive Center.

(3) Why was the role of VOCs in O3 formation and pollution during TC events not analyzed in the study?

*Response:* Thank you for raising this important point regarding the role of photochemical processes in ozone formation. We fully acknowledge that ozone generation fundamentally relies on photochemical reactions involving local emissions

of volatile organic compounds (VOCs) and nitrogen oxides ($NO_x$), particularly in the middle and upper boundary layer. Furthermore, near the surface, ozone is affected by dry deposition and may be titrated by NO during nighttime or under VOC-limited regimes, highlighting the complexity of its chemical lifecycle.

However, this study focuses specifically on a period dominated by the influence of a tropical cyclone (TC). During such synoptic events, meteorological forcing from the TC often surpasses local photochemical processes and becomes the primary driver of rapid and pronounced changes in ozone concentration and distribution — both horizontally and vertically. While the chemical foundation is essential and has been extensively studied in the existing literature, the objective of this paper is to isolate and examine the dynamic and transport mechanisms under such exceptional meteorological conditions, which remain relatively less explored.

Thus, we deliberately centered our analysis on dynamical processes to elucidate how TC-induced circulation reshapes ozone distribution. This focus does not imply that chemical processes are unimportant; rather, it aims to highlight the dominant role of synoptic-scale dynamics in driving short-term ozone variability in the present case. We believe this approach helps clarify the distinctive nature of our case study — namely, how a TC can primarily modulate ozone variations through physical transport during its passage. The manuscript has been revised accordingly (Lines 666 - 668):

In addition, studies have demonstrated that chemical processes also play a significant role in variations in ozone concentration. These include biogenic volatile organic compounds (BVOCs) that serve as precursors for ozone formation, elevated temperatures, enhanced solar radiation, and increased relative humidity in the peripheral regions of typhoons, creating favorable conditions for ozone production. Furthermore, the interaction between anthropogenic and biogenic sources can accelerate ozone formation (Wang et al., 2022). Although these processes have been systematically examined in the existing literature, the present study focuses primarily on the dynamic processes of ozone transport during typhoon events. We intend to further explore the underlying mechanisms in subsequent research.

(4) What are the spatial resolutions of all datasets used in the research?

*Response:* Thank you for your suggestion. The datasets used in this study include the FNL data at a 1°×1° resolution, the MEIC inventory data at 0.25°×0.25°, and the MEGAN biogenic emission data at 0.5°×0.5°. Due to the uneven spatial distribution of other observational data, a specific resolution cannot be provided for them. The relevant descriptions in the manuscript have been revised to present this information more clearly (Lines 257 - 258):

Biogenic VOCs emissions **at a 0.5°×0.5° spatial resolution** are sourced from the Megan emission inventory.

Minor comments:

(5) Page 4, Lines 110 – 112: The detection heights of the wind profiler radar differ across the three stations. Could this difference affect the vertical profile data?

*Response:* Thank you for your question. The wording in the original text was not precise; it should have stated that while the radar detection systems have varying blind zones, their overall detection ranges are sufficient to cover both the boundary layer and the layers above it. This does not affect the conclusions drawn in the manuscript. The relevant text has been revised accordingly (Lines 137 - 138):

The **blind zones** of the wind profiler radar is 100 meters in HD and GZ, while it is 300 meters in HK.

(6) Page 4, Line 119: The temporal resolution of the best-track TC dataset is 6 hours. Could you clarify how this dataset is matched with the 1-hour resolution data used in the study?

*Response:* Thank you for your suggestion. We have provided a correspondence table between time and distance in Table 1. In cases where a direct correspondence was not available in the text, linear interpolation was applied to calculate the distance.

(7) Page 4, Line 120: Please provide references defining "TC days."

*Response:* Thank you for your suggestion. The relevant literature citations have been added to the manuscript (Lines 151 - 152):

In this study, the TC days were defined as when the TC track enters the region defined by 10°N-30°N latitude and 100°E-130°E longitude **(Zhang et al., 2024)**.

(8) Page 5, Line 131: Please provide a reference for the national standard of "MDA8", e.g., doi: 10.1016/j.rse.2021.112775

*Response:* Thank you for your suggestion. The relevant citation has been added to the text (Lines 164 – 166):

According to the national standard in China, the maximum daily 8-hour average $O_3$ concentration (MDA8) should not exceed 160 µg/m³ **(https://www.mee.gov.cn/ywgz/fgbz/bz/bzwb/dqhjbh/dqhjzlbz/201203/t20120302_224165.shtml)**.

(9) Page 5, Line 132: Please provide references that define "O3 pollution" in the GBA.

*Response:* Thank you for your suggestion. This definition was formulated by analogy with the criteria for heatwaves adopted in several previous studies. The relevant description in the text has been revised to clarify the source of the definition (Lines 168 – 171).

The criterion for $O_3$ pollution in the GBA is defined as occurring when more than one-third of the selected stations record concentrations exceeding the threshold, **drawing on an analogous approach used in previous studies to define regional heatwave events in this area (Zhang et al., 2024).**

(10) Table 2: Please include the calculation methods for the statistical metrics, such as MB, FE, and FB.

*Response:* Thank you for your suggestion. The formulas corresponding to these parameters have been added (Lines 275 – 282):

The calculation formula for the parameters is as follows:

$$\text{RMSE} = \sqrt{\frac{1}{N}\sum_{i=1}^{N}(F_i - O_i)^2}, (3)$$

$$\text{MB} = \frac{1}{N}\sum_{i=1}^{N}(F_i - O_i)^2, (4)$$

$$FB = \frac{\bar{F} - \bar{O}}{0.5 \times (\bar{F} + \bar{O})}, (5)$$

$$FE = \frac{\sum_{i=1}^{N} (F_i - \bar{F})^2}{\sum_{i=1}^{N} (O_i - \bar{O})^2} - 1, (6)$$

$$IOA = 1 - \frac{\frac{1}{N}\sum_{i=1}^{N} (F_i - O_i)^2}{\frac{1}{N}\sum_{i=1}^{N} (|F_i - \bar{O}| + |O_i - \bar{O}|)^2}, (7)$$

Note: $F$: Simulated value; $\bar{F}$: Mean of simulated values; $O$: Observed value; $\bar{O}$: Mean of observed values; $N$: Sample size.

(11) Page 8, Line 201: Please clarify the standard used to define "O3 pollution day" and cite related references, for example: doi:10.1016/j.rse.2024.114482

*Response:* Thank you for your suggestion. Based on the MDA8 standard explained above, we define days exceeding this threshold as ozone pollution days.

(12) Page 9, Lines 220–221: Please provide possible explanations for the observed results.

*Response:* Thank you for your suggestion. Higher latitude regions generally provide photochemical environments that are more conducive to ozone formation, especially during summer, while complex topography in these areas further facilitates ozone accumulation. We have added relevant explanations in the revised manuscript (Lines 312 – 315):

Cities located at higher latitudes, often with mountainous landscapes, generally exhibited higher $O_3$ concentrations, a pattern that may be attributed to photochemical environments more conducive to ozone formation as well as terrain features favoring ozone accumulation. In contrast, lower-latitude cities tended to have lower ozone levels.

(13) Figure 3: Some regions in the figure appear to lack data. Please clarify.

*Response:* Thank you for your comment. This discrepancy arises from gaps in the ground-based ozone observations, which do not affect the validity of the conclusions drawn in our study.

(14) Page 13, Line 239: It appears that subfigure (d) is missing from the figure.

*Response:* Thank you for your suggestion. This reference should correspond to Figure 2b, and the correction has been made in the manuscript.

(15) Figure 10: The units of O3 in the text are μg/m3, but they are given in ppmv here.

Please maintain consistency in the units.

*Response:* Thank you for your suggestion. The figure has been redrawn after unit conversion to ensure consistency.

(16) Figure 12: Why do the VWS grids differ in size among the three station regions?

Please clarify this discrepancy.

*Response:* Thank you for your question. This discrepancy arises from variations in the observational data resolution among different wind profile radars, which results in the differences in resolution shown in the figure.